# 🚲 Bisecle: Binding and Separation in Continual Learning for Video Language Understanding

**Yue Tan**
School of Computer Science
University of New South Wales
Sydney, Australia
yue.tan@unsw.edu.au

**Xiaoqian Hu**
School of Computer Science
University of New South Wales
Sydney, Australia
xiaoqian.hu@student.unsw.edu.au

**Hao Xue**
School of Computer Science and Engineering
University of New South Wales
Sydney, Australia
hao.xue1@unsw.edu.au

**Celso De Melo**
DEVCOM Army Research Laboratory
USA
celso.miguel.de.melo@gmail.com

**Flora D. Salim** *
School of Computer Science and Engineering
University of New South Wales
Sydney, Australia
flora.salim@unsw.edu.au

## Abstract

Frontier vision-language models (VLMs) have made remarkable improvements in video understanding tasks. However, real-world videos typically exist as continuously evolving data streams (e.g., dynamic scenes captured by wearable glasses), necessitating models to continually adapt to shifting data distributions and novel scenarios. Considering the prohibitive computational costs of fine-tuning models on new tasks, usually, a small subset of parameters is updated while the bulk of the model remains frozen. This poses new challenges to existing continual learning frameworks in the context of large multimodal foundation models, i.e., catastrophic forgetting and update conflict. While the foundation models struggle with parameter-efficient continual learning, the hippocampus in the human brain has evolved highly efficient mechanisms for memory formation and consolidation. Inspired by the rapid **Bi**nding and pattern **se**paration mechanisms in the hippocampus, in this work, we propose **Bisecle** for video-language **c**ontinual **le**arning, where a multi-directional supervision module is used to capture more cross-modal relationships and a contrastive prompt learning scheme is designed to isolate task-specific knowledge to facilitate efficient memory storage. Binding and separation processes further strengthen the ability of VLMs to retain complex experiences, enabling robust and efficient continual learning in video understanding tasks. We perform a thorough evaluation of the proposed Bisecle, demonstrating its ability to mitigate forgetting and enhance cross-task generalization on several VideoQA benchmarks.

---

*Corresponding Author.

39th Conference on Neural Information Processing Systems (NeurIPS 2025).

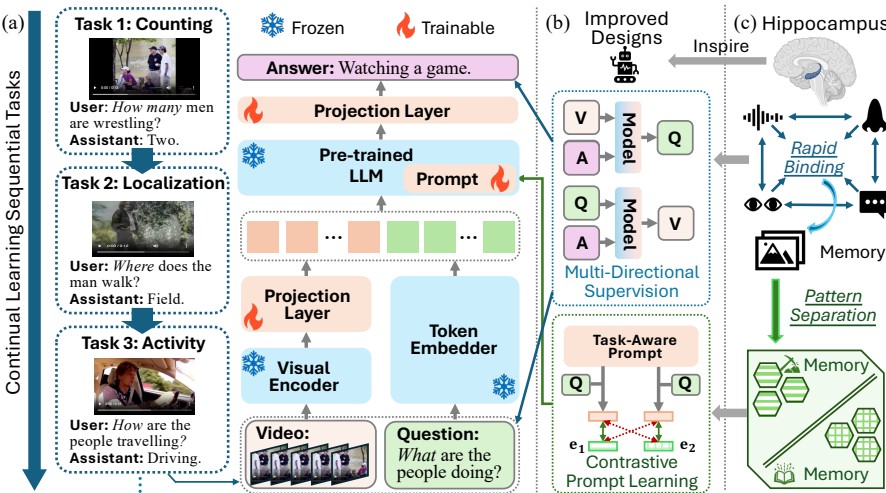

Figure 1: (a) The backbone of our continual learning framework for video understanding; (b) The improved designs proposed in this paper; (c) The motivations from a neurobiological perspective.

# 1 Introduction

Recent advances in large multimodal foundation models, such as vision-language models (VLMs) combining vision encoders with large language models (LLMs), have demonstrated remarkable capabilities in video understanding tasks like visual question answering (VideoQA) and video captioning [1, 2, 3]. However, in real-world applications, these models often face continuously evolving data streams, where the domain and data distribution exhibit inevitable shifts over time [4]. To address this challenge, continual learning has emerged as a critical research direction, enabling models to adapt to new tasks while retaining knowledge from previous ones [5, 6]. While traditional continual learning methods typically optimize the entire model, this approach becomes impractical in the context of large multimodal foundation models due to their massive scale and the prohibitive computational costs of fine-tuning the whole framework, including both the visual encoder and LLM [7, 8]. A more feasible alternative is to update only a small subset of pivot parameters while keeping the bulk of the model frozen [9, 10]. Figure 1(a) provides a sketch map of parameter-efficient continual learning for video understanding, where only the lightweight projection layers are learnable during the continual learning process to adapt to sequential tasks [9, 11].

Although updating only a small fraction of parameters can lead to great feasibility and training efficiency, this strategy also introduces significant challenges. ***Challenge 1: catastrophic forgetting***. As the model is continuously trained on data from new tasks, the multimodal associations learned from previous tasks are gradually overwritten or lost. Due to the limited number of trainable parameters, the learnable modules struggle to maintain the knowledge for old tasks, making the forgetting problem more severe compared to full-model fine-tuning scenarios. ***Challenge 2: update conflict***. Given the diversity and heterogeneity of video understanding tasks, they usually require the model parameters to induce incompatible gradient signals. When the trainable parameters are limited, these divergent adaptation needs inevitably lead to conflicts, as the scarce parameters are hard to simultaneously optimize for multiple, often incompatible objectives.

While large multimodal foundation models struggle with parameter-efficient continual learning, our brains have evolved highly efficient mechanisms to tackle similar challenges in real-world scenarios. Responsible for memory formation and consolidation, the hippocampus in the human brain employs several unique mechanisms to handle complex, dynamic information streams [12, 13]. To efficiently encode episodic memories, the hippocampus utilizes a **rapid binding** mechanism to link multimodal information dynamically into cohesive memory [14, 15]. As shown in the upper part of Figure 1(c), the visual, auditory, and contextual cues are integrated through synchronized neural activity, enabling the hippocampus to form multimodal memory traces. Meanwhile, to minimize interference between similar memories, the hippocampus employs **pattern separation** mechanism to generate distinct neural representations for overlapping inputs [16, 17]. As demonstrated in the lower part of Figure 1(c), this process ensures that memories of different events are stored in different

neural subspaces through sparse coding in the dentate gyrus. With efficient mechanisms like rapid binding and pattern separation, the hippocampus can achieve robust memory formation and retrieval using only 0.5% of total brain neurons. Considering the powerful memorizing capability of the hippocampus, a natural question arises: *Can these neurobiological principles inspire the design of parameter-efficient continual learning frameworks for video understanding?*

To answer the above question, in this paper, we propose a simple yet effective method, **Bisecle**, inspired by rapid **Bi**nding and pattern **se**paration mechanisms, for video-language **c**ontinual **le**arning (see Figure 2 for the icon). To be more specific, to address *Challenge 1*, we design a multi-directional supervision module (as shown in the upper part of Figure 1(b)) to mimic the hippocampus's rapid binding capability. Specifically, apart from the task-specific training objective that takes videos and user questions as inputs and expected answers as outputs, we introduce auxiliary reconstruction tasks, including predicting questions from videos and answers and generating videos from questions and answers, to enforce bidirectional binding between modalities. This approach improves the semantic alignment between cross-modal task elements, thereby mitigating catastrophic

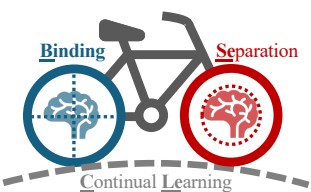

Figure 2: A "Bisecle" with two wheels (binding and separation) running on the road (a sequence of tasks).

forgetting. Moreover, to handle *Challenge 2*, we propose a contrastive prompt learning module (as shown in the lower part of Figure 1(b)) inspired by the pattern separation mechanism. Concretely, we introduce task-specific task type embeddings to store information for different tasks and leverage them to optimize the learnable prompts using a contrastive loss. This loss enhances the agreement between reweighted prompts obtained by attending to task-specific questions and their corresponding learnable task-specific knowledge, which alleviates the update conflict in shared parameters. In this way, Bisecle effectively isolates task-specific knowledge and reduces the potential update conflict across different tasks during the continual learning process. To sum up, the contributions of this paper are three-fold:

- **Neurobiology-Inspired Problem Reframing.** We uniquely reframe this challenge through the lens of neurobiological mechanisms (i.e., rapid binding and pattern separation) to address catastrophic forgetting and update conflicts under strict parameter efficiency constraints.

- **Novel Methodology.** We propose a novel method, Bisecle, that integrates multi-directional supervision and contrastive prompt learning to enable robust and efficient continual learning in video understanding tasks with multi-modal LLMs.

- **Extensive Experiments.** We conduct extensive experiments to validate the effectiveness of Bisecle, demonstrating significant improvements in mitigating forgetting and enhancing cross-task generalization across three VideoQA benchmarks.

## 2 Related Works

**Continual Learning for Large Language Models.** Continual learning is a promising technique to extend large language models (LLMs) to evolving tasks and applications [18, 19, 20]. Existing methods to achieve continual learning on LLMs can be broadly categorized into three classes: continual pre-training [21, 22, 23, 24], continual instruction tuning [25, 26, 27], and continual alignment [28, 29, 30], each focusing on a specific stage in the deployment of LLMs [31]. Among them, continual instruction tuning most closely aligns with the objective of extending LLMs to video understanding tasks, since it enables the model to adapt to new tasks while retaining previously learned knowledge through task-specific instructions [32, 33, 26]. To address the catastrophic forgetting problem, Contunual-T0 [32] utilizes a memory buffer-based rehearsal mechanism to organize and replay the data of previous tasks. DynaInst [34] introduces dynamic instruction replay and local minima-inducing regularize to enhance the generalizability of models while preserving low computational cost. To enable parameter-efficiency continual learning with LLMs, Progressive Prompts [35] aims to freeze most parameters and only tunes the task-specific prompts for each task in the continual learning process. Jang et al. [36] propose to learn a small expert adapter on top of the LLM for each task and allocate a corresponding expert for each new-coming task. Despite their efficiency in natural language processing tasks, it is non-trivial to apply them to video understanding due to the unique challenges posed by multimodal data, such as the need for robust cross-modal alignment and the dynamic nature of video content.

**Continual Learning for Video Understanding.** To handle different video understanding tasks in a continual learning manner, recent studies explore existing continual learning techniques for video understanding models [4, 37, 38, 39, 40, 41, 42]. As one of the representation methods, CLOVE [37] utilizes a scene graph-based prompt mechanism for data replay in continual learning for videoQA. CL-VQA [38] models the intricate relationships between different modalities via multimodal decoupled prompts. DAM [43] introduces a dynamic merging mechanism for the parameters of adapters, which aims to handle the domain-incremental continual learning problem. In [44], a continual predictive learning approach is developed to learn a mixture world model via predictive experience replay. Tang et al. [6] propose a benchmark that systematically formulates the learning paradigm and provides a comprehensive evaluation of the existing methods. *Nevertheless, the above methods do not leverage LLMs for video understanding, thereby falling short in the comprehension and interpretation of semantic information.* To harness the powerful capability of LLMs for video understanding with evolving tasks, very recently, Cai et al. [9] propose the first LLM-based video understanding framework with continual learning. The proposed method, ColPro, incorporates various prompting techniques (i.e., question constraint, knowledge acquisition, and visual temporal awareness) to deal with the catastrophic forgetting issue. Different from ColPro, our proposed method Bisecle provides a simpler strategy that leverages a multi-directional supervision module and a contrastive prompt learning mechanism to strengthen the memorization process and reduce update conflicts, significantly enhancing the performance in continual video understanding.

## 3 Methodology

### 3.1 Problem Definition and Backbone Architecture

**Problem Definition.** In this paper, we focus on continual learning for video question answering (VideoQA), one of the most representative and essential tasks in video understanding. Consider a sequence of tasks indexed from 1 to $T$. Given the $t$-th task, we have its dataset $\mathcal{D}^t = \{d_1^t, \cdots, d_{n_t}^t\}$, where $n_t$ is the number of data samples and each data sample $d_i^t = < V_i^t, Q_i^t, A_i^t >$ consists of three elements: video $V_i^t$, question $Q_i^t$, and answer $A_i^t$. The objective is to train a video-language model $f(V_i^t, Q_i^t) = A_i^t$ on the datasets of the sequence of tasks, and the model can achieve strong performance on both current (e.g., the $T$-th task) and previous (e.g., the $1, \cdots, T-1$-th tasks) tasks. In continual learning, different tasks should exhibit a certain level of diversity, reflected in differences in their data distributions [9, 37, 45]. A brief example of task evolving is demonstrated in the left part of Figure 1(a).

**LLM-based Backbone Model.** In this work, we consider a LLM-based video-language model serving as $f(V_i^t, Q_i^t) = A_i^t$ for video understanding tasks. Following Cai et al. [9], we employ the LLaMA-Adapter framework [11] with a ViT [46] visual encoder as our backbone model. As shown in the right part of Figure 1(a), the visual encoder takes the video $V_i^t$ as its input, with a following projection layer to transfer it into a sequence of visual tokens $\mathbf{V}_i^t = \left[\mathbf{v}_1, \ldots, \mathbf{v}_{N_{i,v}^t}\right] \in \mathbb{R}^{N_{i,v}^t \times D}$, where $N_{i,v}^t$ is the number of frames in $V_i^t$, and $D$ denotes the channel dimension for the extracted frame feature. Correspondingly, the question and answer tokens output by the pre-trained fixed tokenizer are denoted as $\mathbf{Q}_i^t = \left[\mathbf{q}_1, \ldots, \mathbf{q}_{N_{i,q}^t}\right] \in \mathbb{R}^{N_{i,q}^t \times D}$ and $\mathbf{A}_i^t = \left[\mathbf{a}_1, \ldots, \mathbf{a}_{N_{i,a}^t}\right] \in \mathbb{R}^{N_{i,a}^t \times D}$, respectively, where $N_{i,q}^t$ and $N_{i,a}^t$ denote the number of question tokens and answer tokens. Taking $\mathbf{V}_i^t$ and $\mathbf{Q}_i^t$ as the initial input, a frozen LLM with a trainable projection layer is expected to predict the answer tokens $\mathbf{A}_i^t$ in an autoregressive fashion. To fine-tune the learnable parameters of model $f$, a cross-entropy loss is employed, which can be written as:

$$\mathcal{L} = -\log P(A_i^t \mid V_i^t, Q_i^t) = -\sum_{k=0}^{N_{i,a}^t - 1} \log P\left(\mathbf{a}_{i,k+1}^t \mid \mathbf{V}_i^t, \mathbf{Q}_i^t, \mathbf{A}_{i,\leq k}^t\right), \tag{1}$$

where $\mathbf{A}_{i,\leq k}^t = [\mathbf{a}_{i,1}^t, \cdots, \mathbf{a}_{i,k}^t]$ includes the first $k$ tokens of the answer sequence. Note that the modules with a large number of parameters (i.e., the LLM and visual encoder) are frozen, leaving only a small number of adaptive modules as learnable. This design ensures parameter-efficient training and adaptation throughout the continual learning process with sequential tasks.

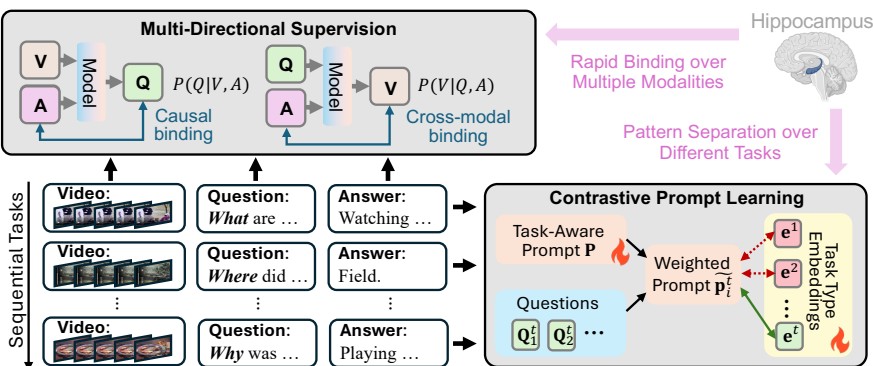

Figure 3: The sketch map to explain the two modules in Bisecle, multi-directional supervision and contrastive prompt learning.

## 3.2 Rapid Binding-Inspired Multi-Directional Supervision

Although the aforementioned backbone model can leverage the reasoning capabilities of LLMs while enabling efficient continual fine-tuning, like other continual learning systems, its performance is still impacted by the issue of **catastrophic forgetting** [9, 47, 48]. Specifically, the model may lose previously learned knowledge, such as the ability to focus on key frames and perform question-specific semantic reasoning in older video understanding tasks, when trained on new tasks, leading to performance degradation. In the context of frozen LLM and visual encoder, this forgetting phenomenon becomes even more severe. This is because the neurons responsible for visual and language understanding are fixed, making it difficult for the model to learn knowledge for scenario-specific understanding required for new tasks. Due to the single training mode where the model is only trained to predict answers given videos and questions, the learnable adapters struggle to capture deeper task understanding, such as the underlying semantic reasoning required for complex video understanding tasks, as well as the intricate cross-modal relationships between visual and textual information. This limitation hinders their ability to adapt effectively across different new tasks, particularly those requiring nuanced multimodal interactions.

**Neurobiological Motivation - Rapid Binding.** Given the similar challenge of memory and knowledge establishment, how does the human brain address this issue? The hippocampus, a critical region for memory formation, employs a rapid binding mechanism to dynamically link multimodal information, such as visual, auditory, and contextual cues, into cohesive episodic memories. This process is facilitated by synchronized neural activity, particularly theta-gamma oscillations, which enable efficient encoding and retrieval of complex associations. Importantly, this binding is **multi-directional**: given a partial cue (e.g., a visual scene), the hippocampus can reconstruct the associated context (e.g., the corresponding event or emotion), and **vice versa**, ensuring robust memory retention even under partial or noisy inputs. By deepening the understanding of multimodal information through these dynamic associations, the hippocampus strengthens memory traces and enhances the ability of brains to retain and recall complex experiences.

**Multi-Directional Supervision.** Inspired by the rapid binding mechanism of the hippocampus [15, 49, 50, 51], we propose a multi-directional supervision module for continual learning of LLM-based video-language models. Considering the VideoQA task, an essential data sample, corresponding to a "memory" for video-language models, is composed of video $V_i^t$, question $Q_i^t$, and answer $A_i^t$. The conventional learning objective is to maximize $\log P(A_i^t \mid V_i^t, Q_i^t)$, i.e., the likelihood of generating the correct answer $A_i^t$ given the video $V_i^t$ and question $Q_i^t$. Despite its alignment with the actual downstream task, the current objective is single-directional. That is to say, the model only learns the unidirectional connection from videos and questions to answers, while ignoring the potential for bidirectional or multi-directional associations, such as the connections from $V_i^t$, $A_i^t$ to $Q_i^t$ and which from $Q_i^t$, $A_i^t$ to $V_i^t$. Such a single-directional supervision may limit the ability of the model to capture deeper causal and cross-modal relationships, thereby preventing it from forming a comprehensive understanding of the current training task. As a result, with insufficient learnable parameters, the model becomes more vulnerable to suffering from catastrophic forgetting.

To address the limitations of single-directional supervision, in Bisecle, we produce a multi-directional supervision module that incorporates multiple directions of learning objectives among video $V_i^t$, question $Q_i^t$, and answer $A_i^t$. Concretely, to explicitly bind the causal relationship between question $Q_i^t$ and its corresponding video $V_i^t$ and answer $A_i^t$, we introduce a question prediction task as auxiliary supervision. Formally, the learning objective function can be written as:

$$\mathcal{L}_Q = -\log P(Q_i^t \mid V_i^t, A_i^t) = -\sum_{k=0}^{N_{i,q}^t - 1} \log P\left(\mathbf{q}_{i,k+1}^t \mid \mathbf{V}_i^t, \mathbf{A}_i^t, \mathbf{Q}_{i,\leq k}^t\right), \qquad (2)$$

where $\mathbf{Q}_{i,\leq k}^t = [\mathbf{q}_{i,1}^t, \cdots, \mathbf{q}_{i,k}^t]$ includes the first $k$ tokens of the question sequence. Through the question prediction loss $\mathcal{L}_Q$, the model can capture more causal knowledge about the specific VideoQA task. For example, when given a video of a car accident and the answer "brake failure", the model can infer the question "what caused the accident?", which enables the model to establish a deeper understanding of the causal relationship between the question and the answer.

Apart from the question prediction task, Bisecle also incorporates a video prediction task to provide extra supervision signals. Through predicting the video sequence based on the question and answer, the video-language model can enhance its ability to capture cross-modal relationships as well as temporal dynamics, leading to a more robust understanding of the task content. However, it is challenging to directly predict the visual tokens as they are out of the discrete token space of language models. Inspired by [52], we adopt an alternative strategy that maximizes the mutual information between the input visual tokens of frames and the output feature of LLMs. Specifically, the learning objective function of our video prediction task can be written as:

$$\begin{aligned}
\mathcal{L}_V = -\log P(V_i^t \mid Q_i^t, A_i^t) &= -\sum_{k=0}^{N_{i,v}^t - 1} \log P\left(\mathbf{v}_{i,k+1}^t \mid \mathbf{Q}_i^t, \mathbf{A}_i^t, \mathbf{V}_{i,\leq k}^t\right) \\
&= -\sum_{k=0}^{N_{i,v}^t - 1} \log \frac{\exp\left(\mathbf{v}_{i,k+1}^{t\top} \mathbf{h}_{i,k}^t\right)}{\sum_{j=1}^{N_{i,v}^t} \exp\left(\mathbf{v}_{i,j}^{t\top} \mathbf{h}_{i,k}^t\right)},
\end{aligned} \qquad (3)$$

where $\mathbf{h}_{i,k}^t$ is the output token representation of LLMs before the start of visual tokens. The video prediction loss encourages the model to predict the sequence of video frames based on the preceding frames, thereby enhancing its ability to capture temporal dependencies and improve cross-modal video understanding within the current training task in the continual learning process.

### 3.3 Pattern Separation-Inspired Contrastive Prompt Learning

Apart from the above challenge, another critical issue is the **update conflict** of the learnable parameters. To be concrete, when multiple tasks compete for updates to the same set of parameters, the model may prioritize new tasks at the expense of previously learned knowledge. Especially when the learnable parameters are scarce, this conflict becomes more pronounced, leading to significant interference between tasks. While existing approaches employ task-specific prompts [9, 53] to mitigate this issue, they require precise matching between test-time questions and training tasks to select the corresponding prompts. This matching process proves particularly challenging in open-world QA scenarios, where additional knowledge is often needed to identify the correct task association. Moreover, the substantial divergence between task-specific prompts may lead to overfitting to individual tasks, consequently degrading model performance. Therefore, how to alleviate the update conflict issue without introducing task-specific parameters remains a significant challenge.

**Neurobiological Motivation - Pattern Separation.** From the perspective of neurobiology, the memory encoding mechanism of our brains can provide a promising solution to deal with the update conflict issue. The dentate gyrus, a critical region in the hippocampus for higher-order cognitive functions, employs a pattern separation mechanism to encode distinct representations for overlapping inputs. This process is facilitated by sparse coding and lateral inhibition, which ensure that similar but non-identical memories are stored in **non-overlapping neural subspaces**. Notably, this mechanism allows the brain to differentiate between similar experiences while preserving the unique details of

each knowledge. By isolating task-specific knowledge through pattern separation, the hippocampus can efficiently retain and recall complex experiences without interference.

**Contrastive Prompt Learning.** To enable the model to learn task-specific knowledge while maintaining model generalization ability and parameter efficiency, we introduce task-aware learnable prompts that shared across all training tasks and then introduce a contrastive prompt learning strategy to optimize the prompts with task-specific restrictions. Formally, the task-aware learnable prompts are attached to multiple transformer layers in the LLM as adaptive task adapters, which can be denoted as a prompt matrix $\mathbf{P} \in \mathbb{R}^{(N_p \times L_p) \times D}$, where $N_p$ is the number of learnable prompt tokens at each layer, $L_p$ is the number of transformer layers where prompts are injected, and $D$ is the feature dimension of LLMs. Here, $N_p$ and $L_p$ are hyper-parameters to define the size of prompts. At the corresponding layer, the $N_p$ learnable prompt embeddings are concatenated with the question embeddings.

While shared prompts help preserve generalization ability across diverse and open-set downstream tasks during testing, they inevitably suffer from knowledge interference among different training tasks during prompt parameter updates. To mitigate this issue, we draw inspiration from the pattern separation mechanism and propose a contrastive prompt learning strategy. This approach regularizes learnable prompts with task-specific embeddings, explicitly strengthening their associations with task-specific knowledge. Specifically, we allocate a task type embedding $\mathbf{e}^t \in \mathbb{R}^D$ for each task $t$, serving as the non-overlapping neural subspace to store the knowledge for the specific task. We dynamically reweight the learnable prompts based on specific training questions and employ a contrastive loss to enforce mutual agreement between each reweighted prompt and its corresponding task type embedding. This approach enables task-aware prompts to adapt to both the specific tasks and questions through explicit alignment with task-related knowledge, thereby mitigating potential conflicts in these bottleneck parameters.

In formal, given a question token matrix $\mathbf{Q}_i^t$ and the task-aware prompt matrix $\mathbf{P}$, the reweighted prompt vector can be calculated by $\widetilde{\mathbf{p}}_i^t \in \mathbb{R}^D = (\mathbf{q}_i^t \cdot \mathbf{P}^\top) \cdot \mathbf{P}$, where $\mathbf{q}_i^t \in \mathbb{R}^D$ is the averaged question representation obtained by mean-pooling the token embeddings of $\mathbf{Q}_i^t$ along the sequence dimension. In this way, $\widetilde{\mathbf{p}}_i^t$ effectively encodes task-specific knowledge with the awareness of the current input context. Notably, more complicated mechanisms (e.g., attention across multi-layer outputs) can be applied to calculating $\widetilde{\mathbf{p}}_i^t$, but we empirically found that our lightweight reweighting achieves comparable performance with reduced computational overhead. Once we obtain $\widetilde{\mathbf{p}}_i^t$, the contrastive loss across $\widetilde{\mathbf{p}}_i^t$ and $\mathbf{e}^t$ can be calculated by:

$$\mathcal{L}_P = -\log \frac{\exp\left(\widetilde{\mathbf{p}}_i^t \cdot \mathbf{e}^t / \tau\right)}{\sum_{t' \in \mathcal{A}(t)} \exp\left(\widetilde{\mathbf{p}}_i^t \cdot \mathbf{e}^{t'} / \tau\right)}, \tag{4}$$

where $\mathcal{A}(t) = \{0, \cdots, t\}$ is the set denoting the index of previous and current tasks from $0$ to $t$, and $\tau$ is the temperature parameter that can adjust the tolerance for feature difference. Through this contrastive mechanism, learnable prompts are discriminately aligned with the task-specific embeddings, mitigating the problem of inter-task interference and further alleviating update conflicts during continual learning. In this mechanism, the task type embedding matrix $\mathbf{e}^t$ functions analogously to the dentate gyrus in human memory systems, strategically partitioning different task representations into distinct latent regions. With a trade-off hyper-parameter $\gamma$, the overall training loss of Bisecle can be written as $\mathcal{L} = \mathcal{L}_A + \mathcal{L}_Q + \mathcal{L}_V + \gamma \mathcal{L}_P$.

## 4 Experiments

### 4.1 Experimental Setup

**Datasets.** We conduct experiments on three VideoQA datasets, i.e., NExT-QA [54], DramaQA [55], and STAR [56]. For NExT-QA, we split questions into eight task types (e.g., causal why/how, temporal what/when, and descriptive where/how many). Following prior work [9], we adopt the task order <TP, CW, DC, TC, DL, DO, TN, CH>. For DramaQA, we partition questions into five types and use the order with maximum forgetting, i.e., <What, Who, Where, How, Why>. For STAR, we follow its reasoning tasks <Interaction, Sequence, Prediction, Feasibility> to evaluate situational understanding in the continual learning scenarios. More details are in Appendix A.1.1.

Table 1: Comparison with state-of-the-art methods on NExT-QA, DramaQA, and STAR datasets. **Bold** and underline represent the best and runner-up results in each column.

| Method | NExT-QA | | DramaQA | | STAR | |
|---|---|---|---|---|---|---|
| | Acc ($\uparrow$) | Fog ($\downarrow$) | Acc ($\uparrow$) | Fog ($\downarrow$) | Acc ($\uparrow$) | Fog ($\downarrow$) |
| Backbone (LLaMA-Adapter) | 46.58 | 13.83 | 60.99 | 24.39 | 46.89 | 11.54 |
| L2P [53] | 48.82 | 12.25 | 62.50 | 20.67 | 48.25 | 10.82 |
| DualPrompt [57] | 50.62 | 11.74 | 65.89 | 17.93 | 49.73 | 10.15 |
| LAE [58] | 49.38 | 11.47 | 65.82 | 17.35 | 49.15 | 9.87 |
| DAM [43] | 53.88 | 9.99 | 67.37 | 15.19 | 50.64 | 8.92 |
| ProgPrompt [35] | 53.95 | 10.69 | 67.92 | 14.95 | 51.07 | 8.75 |
| ColPro [9] | 55.14 | 7.43 | 71.24 | 12.64 | 48.67 | 8.13 |
| Bisecle (ours) | **62.37** | **5.34** | **71.49** | **10.37** | **52.16** | **7.60** |

**Baselines.** We compare Bisecle to 6 representative methods for visual/video continual learning, including L2P [53], DualPrompt [57], LAE [58], DAM [43], ProgPrompt [35], and ColPro [9]. Among them, ColPro also employs adapter [11] as its backbone, making it a direct counterpart to our method for a fair comparison. More details for baselines can be found in Appendix A.1.2.

**Evaluation Metrics.** We evaluate the performance of baselines and Bisecle with two commonly-used metrics [9, 38, 53]. To evaluate video understanding performance, we adopt the standard metric of average final accuracy (Avg. Acc) over $T$ tasks for multiple-choice question answering. To evaluate their continual learning capability, we employ average forgetting (Avg. Fog) to quantify catastrophic forgetting, where a smaller value indicates better preservation of previously learned knowledge.

**Implementation Details.** We use LLaMA-Adapter [11] as our backbone model, following [9]. We use the pre-trained LLaMA-2-7B [59] as the LLM and ViT-L/14 [60, 61] as the visual encoder, both of which are fixed during the continual learning process. All models are trained for five epochs with a batch size of 32 on all datasets. The number of adapter layers is set to 32, the adapter length is 10, and the weight decay is $0.14$. We conduct all experiments on two NVIDIA H100 GPUs. Detailed experimental settings can be found in Appendix A.

## 4.2 Experimental Results

**Performance Comparison.** The performance comparison on three benchmark datasets is demonstrated in Table 1, from which we have the following observations. ❶ Our method establishes new state-of-the-art results across all datasets, surpassing the backbone model in both accuracy (+15.79% on NExT-QA) and forgetting reduction (8.49% lower Fog on NExT-QA). The superior performance demonstrates the powerful capability to handle continual learning challenges and tackle catastrophic forgetting issues. ❷ The poor performance of the backbone (adapter) highlights its vulnerability to sequential learning, with forgetting rates up to 24.39% on DramaQA, underscoring the need for dedicated continual learning mechanisms for LLM-based video understanding models. ❸ While ColPro employs a more complicated prompt learning mechanism based on the same backbone, our approach achieves superior performance (↑2.35% Acc, ↓2.27% Fog on DramaQA) through lightweight multi-directional supervision and contrastive prompt learning mechanisms.

**Ablation Studies.** To investigate the contribution of each component in our method, we compare Bisecle with different variants with different removed loss terms. According to the results in Table 2, we have the below findings. ❶ The complete version of Bisecle utilizing all loss components demonstrates the best performance across all three datasets. This indicates synergistic effects between the different mechanisms (i.e., multi-directional supervision and contrastive prompt learning), where joint optimization maximizes model capability while minimizing catastrophic forgetting and alleviating update conflict. ❷ In most cases, all loss terms bring positive effects to the performance, indicating the effectiveness of the proposed mechanisms. ❸ Among the three loss terms, $\mathcal{L}_Q$ provides the most significant performance improvement, demonstrating that strengthening the causal relationship between questions and answers is critical for mitigating catastrophic forgetting.

**Data Efficiency.** To verify the performance of Bisecle under data-scarce scenarios, we conduct this experiment under varying training data sizes, as shown in Figure 4. The results demonstrate three key

Table 2: Ablation studies on the effects of different loss components across three datasets.

| Components | | | NExT-QA | | DramaQA | | STAR | |
|:---:|:---:|:---:|:---:|:---:|:---:|:---:|:---:|:---:|
| $\mathcal{L}_Q$ | $\mathcal{L}_V$ | $\mathcal{L}_P$ | Acc (↑) | Fog (↓) | Acc (↑) | Fog (↓) | Acc (↑) | Fog (↓) |
| ✗ | ✗ | ✗ | 46.58 | 13.83 | 60.99 | 24.39 | 46.89 | 11.54 |
| ✓ | ✗ | ✗ | 61.13 | 6.63 | 71.40 | 10.90 | 49.71 | 9.43 |
| ✗ | ✓ | ✗ | 53.94 | 9.61 | 67.39 | 17.15 | 49.48 | 8.86 |
| ✗ | ✗ | ✓ | 55.82 | 8.93 | 64.74 | 19.01 | 47.93 | 9.19 |
| ✓ | ✓ | ✗ | 59.78 | 6.58 | 70.25 | 12.65 | 48.50 | 9.24 |
| ✗ | ✓ | ✓ | 51.95 | 11.73 | 67.84 | 14.10 | 49.68 | 7.97 |
| ✓ | ✗ | ✓ | 61.38 | 7.20 | 68.47 | 14.63 | 51.96 | 8.43 |
| ✓ | ✓ | ✓ | **62.37** | **5.34** | **71.49** | **10.37** | **52.16** | **7.60** |

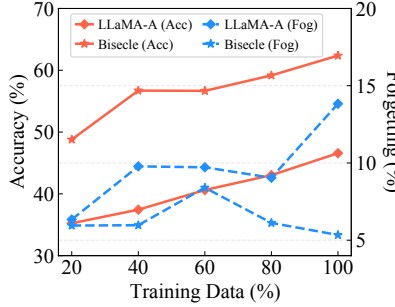

Figure 4: Performance on different sizes of training datasets.

Table 3: Performance on various sizes of LLMs (i.e., LLaMA-3.2-1B, LLaMA-2-7B, and LLaMA-2-13B).

| Method | LLaMA | # of L. Par. | Acc (↑) | Fog (↓) |
|:---|:---:|:---:|:---:|:---:|
| Backbone | 1B | 1,673,890 | 20.39 | 0.10 |
| | 7B | 4,499,456 | 52.81 | 4.69 |
| | 13B | 6,034,560 | 57.98 | 4.52 |
| Bisecle | 1B | 1,777,920 | 21.17 | 0.00 |
| | 7B | 4,540,416 | 53.45 | 2.60 |
| | 13B | 6,085,760 | 59.43 | 3.03 |

findings: ❶ Our method Bisecle consistently outperforms the backbone (i.e., Adapter) across two metrics, indicating its strong robustness when training data is limited. ❷ Bisecle exhibits remarkable resistance to forgetting when sufficient training data is available, indicating the effectiveness of its key mechanisms, i.e., multi-directional supervision and contrastive prompt learning. ❸ Bisecle can achieve superior performance even in low-resource settings, indicating its data efficiency.

**Robustness with Different LLMs.** To validate the adaptation capability of Bisecle with different LLMs, we conduct experiments with LLaMA variants (1B/7B/13B parameters), and the results are shown in Table 3. For the sake of time, we only run experiments on three typical question types of NExT-QA dataset. We have these key observations emerge: ❶ Bisecle achieves accuracy improvements across all model sizes, demonstrating its robustness to LLM backbones. ❷ The forgetting rate is significantly reduced by Bisecle, especially for mid-scale models. This suggests our neurobiologically-inspired designs effectively mitigate catastrophic forgetting regardless of model scale. ❸ Despite adding only a few extra learnable parameters (104k–51k), Bisecle delivers disproportionate benefits.

Table 4: Performance with different numbers of prompt layers.

| # P. Layer | Acc (↑) | Fog (↓) |
|:---:|:---:|:---:|
| 8 | 53.86 | 10.13 |
| 16 | 58.18 | 7.52 |
| 24 | 57.30 | 7.99 |
| 32 | 62.37 | 5.34 |

**Sensitivity to the Numbers of Layers with Prompt Injection.** To study the impact of different numbers of learnable prompts, we vary the numbers of layers with prompt injection (# P. Layer) from 8 to 32, with the results shown in Table 4. The results demonstrate a clear scaling trend: increasing the number of learnable prompt layers usually improves both accuracy and forgetting resistance. This indicates that task-specific parameters play dual critical roles in continual learning with video-language models.

Table 5: Performance of different contrastive prompt learning manners.

| Method | Acc (↑) | Fog (↓) |
|---|---|---|
| Variant 1 | 58.96 | 6.56 |
| Variant 2 | 59.96 | 7.58 |
| Bisecle (ours) | **62.37** | **5.34** |

**Performance of Varying Contrastive Prompt Learning Manners.** Table 5 shows the performance of using different contrastive prompt learning manners on NExT-QA dataset. In variant 1, task type embeddings are computed as the mean of question tokens within the same class rather than as learnable embeddings. In variant 2, task type embeddings are directly used as the contrastive components without the prompt reweighting procedure. It can be observed that by defining the task type embedding as learnable variables, VLMs can better enhance performance on various tasks.

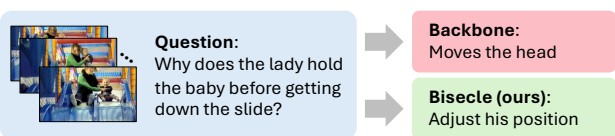

Figure 5: Answers by Backbone and Bisecle.

**Case Study.** We further evaluate the effectiveness of Bisecle by examining failure cases (See Figure 5, more can be found in Appendix B.3) of the backbone model where our approach succeeds. While the backbone model fails to establish causal relationships between questions and answers, Bisecle successfully resolves the case, thanks to the causal understanding capability acquired from multi-directional supervision.

# 5 Conclusion

In this paper, we present Bisecle, a neurobiologically inspired framework for video-language continual learning that addresses critical challenges in adapting large VLMs to dynamic real-world scenarios. By emulating the binding and pattern separation mechanisms in human brain through multi-directional supervision and contrastive prompt learning, Bisecle effectively mitigates catastrophic forgetting while maintaining parameter efficiency. It not only advances the state of continual learning for multimodal systems but also provides new insights into biologically inspired AI design. Extensive evaluations on VideoQA benchmarks demonstrate the superior ability of our method to preserve past knowledge and generalize to novel tasks compared to existing approaches.

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

# A  Experimental Details

## A.1  Detailed Setting of Table 1

### A.1.1  Task and Data Setting

For NExT-QA and DramaQA datasets, we follow the continual learning setting in ColPro [9]. For NExT-QA, there are eight tasks including causal questions (CW, CH), temporal questions (TC, TN, TP), descriptive questions (DC, DL), and other types of questions (DO). For DramaQA, there are five tasks including *what*, *who*, *where*, *how*, and *why*. For STAR, there are four tasks corresponding to *interaction*, *sequence*, *prediction*, and *feasibility*. Table 6, Table 7, and Table 8 give several question examples for each task in NExT-QA, DramaQA, and STAR, respectively.

Table 6: Question examples of different tasks in NExT-QA.

| Task Type | Question Examples |
|---|---|
| TP | What did the man in grey do before the plane took off? 
 What did the man on the stage do before sitting? |
| CW | Why does the man have to throw the plane first in the middle of the video? 
 Why did the man wear hat while riding the horse? |
| DC | How many children are in the video? 
 How many people are cycling? |
| TC | What did the lady do when they reached the bush? 
 What does the man in white do when he walks onto the stage? |
| DL | Where is the boy projecting his photos on? 
 Where is the cat lying? |
| DO | What are the different colors of the balls floating in the pool? 
 What is the possible relationship between the girl and the boy? |
| TN | How did the two women react after the woman in stripes released their hand? 
 How did the dog react after the lady caress it? |
| CH | How did the boy in stripped open the book to see its contents? 
 How did the child use his hands to show his excitement at the start? |

Table 7: Question examples of different tasks in DramaQA.

| Task Type | Question Examples |
|---|---|
| TW | What is being cooked in the pot? 
 What kind of thing is to eat on the plate? |
| DO | Who enters the house with wearing a bag? 
 Who is yelling? |
| DL | Where is Deogi looking while talking? 
 Where did Sukyung put her hand? |
| CH | How does Jiya think will happen if Jiya doesn't come back with the lost money? 
 How did Taejin start to play the game when Taejin is with chairman? |
| CW | Why was Deogi in the kitchen? 
 Why did Haeyoung1 light a fire under the iron pot? |

Table 8: Question examples of different tasks in STAR.

| Task Type | Question Examples |
|---|---|
| Interaction | Which object was eaten by the person? 
 Which object was put down by the person? |
| Sequence | Which object did the person eat after they put down the book? 
 Which object did the person open after they sat at the table? |
| Prediction | What will the person do next? 
 Which object would the person put down next after they take the bag? |
| Feasibility | What else is the person able to do with the dish? 
 Which object is the person able to throw after walking through the doorway? |

### A.1.2  Details of Baseline Methods

*L2P* [53] reframes continual learning as a prompt-selection problem. It maintains a small pool of learnable prompts, each paired with a key, and keeps a large pre-trained transformer frozen. For each incoming sample, L2P computes a query from the input, retrieves the top-N matching prompts, prepends them to the input embeddings, and then updates only the prompt pool and a lightweight classifier via a combined cross-entropy and key-matching loss. This design decouples task-specific from shared knowledge, requires no rehearsal buffer or task IDs at test time, and consistently outperforms state-of-the-art methods across class-incremental, domain-incremental, and task-agnostic benchmarks.

*DualPrompt* [57] is a rehearsal-free continual learning framework that enables a frozen, pretrained vision transformer to learn a sequence of class-incremental tasks without storing any past data. It does so by introducing two small, complementary sets of learnable prompts (G-Prompts for capturing task-invariant "general" instructions and E-Prompts for encoding task-specific "expert" instructions), which are attached to selected multi-head self-attention layers. A simple matching mechanism retrieves the appropriate E-Prompt at test time, and the model is trained end-to-end with a combination of classification and prompt-matching losses.

*LAE* [58] introduces a unified approach for continual learning by leveraging parameter-efficient tuning methods, such as Adapter, LoRA, and Prefix, to adapt pre-trained models to new tasks efficiently. It consists of three key components: 1) learning with calibrated adaptation speeds to align different tuning methods, 2) accumulation of task-specific knowledge into an offline fine-tuning module via momentum updates, and 3) ensemble of online and offline modules during inference to balance performance across old and new tasks. This design ensures robust continual learning performance while minimizing catastrophic forgetting and computational overhead.

*DAM* [43] is a parameter-efficient method designed for continual video question answering. It mitigates catastrophic forgetting by training dataset-specific adapters while dynamically merging them during inference to handle unknown domains and enable knowledge sharing. DAM has demonstrated the effectiveness across diverse video and image tasks.

*ProgPrompt* [35] introduces a novel continual learning approach for language models by progressively learning and concatenating soft prompts for each new task while keeping the base model frozen. This method leverages task-specific prompts to prevent catastrophic forgetting and enables forward transfer by reusing knowledge from previous prompts. Additionally, it incorporates a residual MLP-based reparameterization technique to stabilize training and improve performance, achieving significant gains over existing methods in both few-shot and full-data settings.

*ColPro* [9] is a rehearsal-free continual learning framework that leverages a frozen large language model and three complementary prompting strategies: task-specific question constraint prompting, knowledge acquisition prompting, and visual temporal awareness prompting. These strategies work together to encode question context, multimodal knowledge, and temporal dynamics into prompts that steer the model to learn new tasks without overwriting prior knowledge.

### A.1.3 Model Architecture

The whole model involves one LLM backbone which is the LLaMA-Adapter [11], one visual encoder which is ViT-L/14 [60, 61], a projection layer aligning the output latent representations of different modalities, and another projection layer as the final linear layer that maps the latent representations to logits over the vocabulary space.

For the LLM part, we have chosen LLaMA-7B as the backbone for most experiments except for the implementation to verify the robustness with different sizes of LLMs. We adopt LLaMA-Adapter as the parameter-efficient fine-tuning method. Concretely, instead of updating the full 7B parameters, LLaMA-Adapter freezes the pre-trained LLaMA and only learns the adaptation prompts with 1.2M parameters on top.

For the visual encoder, the input resolution is $224 \times 224$. For each input image, the encoder splits it into 14×14 non-overlapping patches, and each patch is flattened and linearly projected to $D = 1024$ dimensions. The visual encoder consists of 24 transformer encoder layers, each employing multi-head self-attention (16 heads) and a feedforward MLP with a hidden dimension of 4096 (GELU activation), followed by Layer Normalization (Pre-Norm) for stability.

### A.1.4 Hyperparameters and Training Details

We use dataset-specific batch sizes together with AdamW across all tasks. In particular, for NExT-QA we set the batch size to 32, for DramaQA to 4, and for STAR to 16. All experiments employ the AdamW optimizer with a base learning rate of 0.09. Weight decay is 0.14 for NExT-QA and 0.10 for both DramaQA and STAR. Video inputs consist of 10 frames resized to $224 \times 224$, and token sequences are truncated or padded to 128 tokens for NExT-QA, 280 for DramaQA, and 150 for STAR. We train each model for 5 epochs (with 2 warm-up epochs) and fix the random seed to 0 for all tasks. Experiments are conducted on two NVIDIA H100 GPUs (94GB of memory per GPU). The GPU-hours for training is around 500 in total.

Table 9 shows the parameters of our contrastive learning setup, governing how question type representations are learned and how negative examples are weighted in the contrastive loss.

Table 9: Training Details of Contrastive Learning.

|  | NextQA | DramaQA | STAR |
|---|---|---|---|
| # Task Types | 8 | 5 | 4 |
| Task Type Embedding Size | [8, 4096] | [5, 4096] | [4, 4096] |
| Negative Temperature | 1.28 | 1.25 | 1.25 |
| Contrastive Loss Weight | 0.15 | 0.10 | 0.10 |

## B  Additional Results

### B.1  Visualization of Prompts across Tasks.

To learn about how the task type embeddings and weighted prompts of different tasks evolve along the continual learning process, we visualize the samples in NExT-QA test set by t-SNE [62]. In Figure 6, the points in different colors refer to representations after fine-tuning on a specific task for one epoch or four epochs. It suggests that, compared with weighted prompts, task type embeddings are more distinguishable in latent space, indicating that they retain more task-specific knowledge. Also, the distribution of weighted prompts is evolving along the task sequence in a more dynamic way, i.e., the positions of different clusters are changing across epochs, because new tasks generally refresh the understanding of the model on old tasks via task-aware prompt updates. This process subsequently affects the weighted prompts through contrastive learning mechanisms.

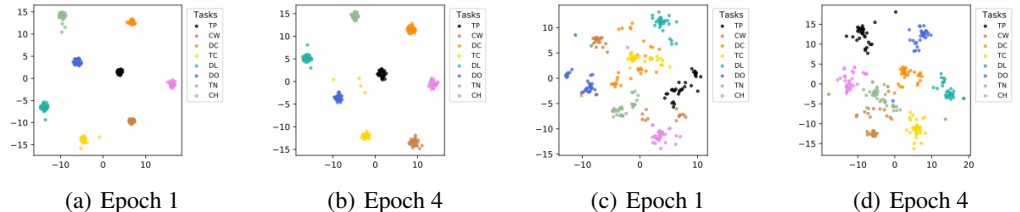

| (a) Epoch 1 | (b) Epoch 4 | (c) Epoch 1 | (d) Epoch 4 |

Figure 6: (a) and (b): t-SNE visualization of learnable *task type* embeddings of different tasks; (c) and (d): t-SNE visualization of *weighted prompts* of different tasks.

## B.2 Performance on Varying Tasks Orders

To investigate how the task order affects the continual learning performance, we evaluate the performance of LLaMA-Adapter and our method across different task orders. Table 10 suggests that the task order can influence both the test accuracy and forgetting rate, due to the various difficulty degrees of the tasks and their intrinsic correlation. It also shows that our method outperforms LLaMA-Adapter in both accuracy and forgetting rate in most cases, demonstrating the stability and robustness of our method to diverse continual learning task settings.

Table 10: Performance of LLaMA-Adapter and Bisecle (ours) across different task orders.

| Task Order | Avg. Acc (↑) | | Avg. Fog (↓) | |
|---|---|---|---|---|
| | LLaMA-A | Bisecle (ours) | LLaMA-A | Bisecle (ours) |
| <CH, DL, TP, TC, DC, DO, TN, CW> | 56.79 | 63.09 | 5.55 | 2.87 |
| <TP, TN, CH, TC, DL, DO, CW, DC> | 57.68 | 57.98 | 5.89 | 7.16 |
| <DO, CW, DC, CH, TP, TC, TN, DL> | 55.63 | 57.95 | 7.15 | 8.93 |
| <CW, DO, TN, DL, TC, TP, DC, CH> | 52.65 | 62.25 | 12.85 | 5.70 |

## B.3 Case Study

We present more cases on NExT-QA and STAR dataset in Figure 7 and Figure 8, respectively, where the backbone fails to give the right answer while our method succeeds. Each subfigure corresponds to a specific task in the learning sequence, helping the readers better learn about the video QA tasks we are working on.

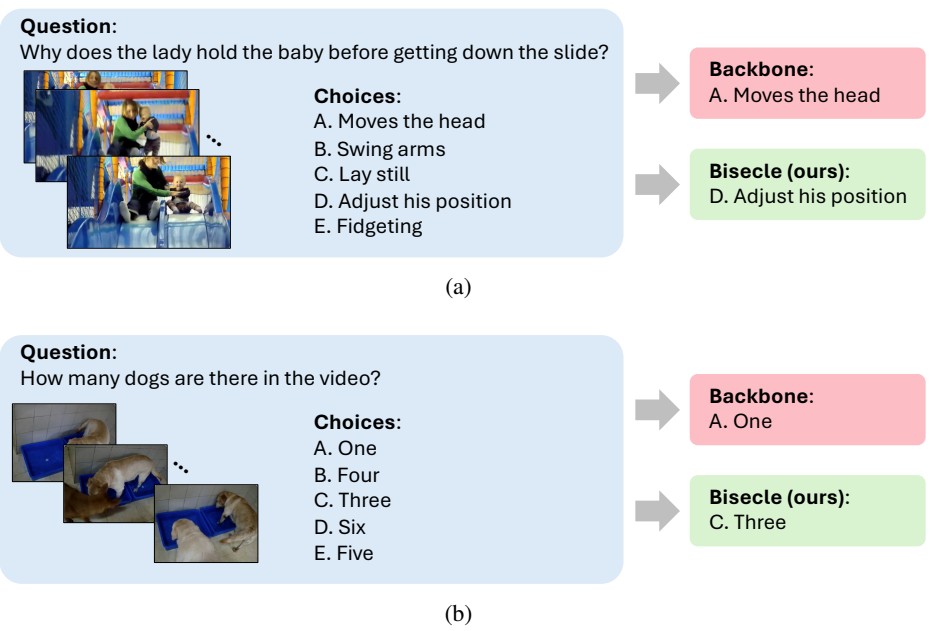

Figure 7: More cases on NExT-QA.

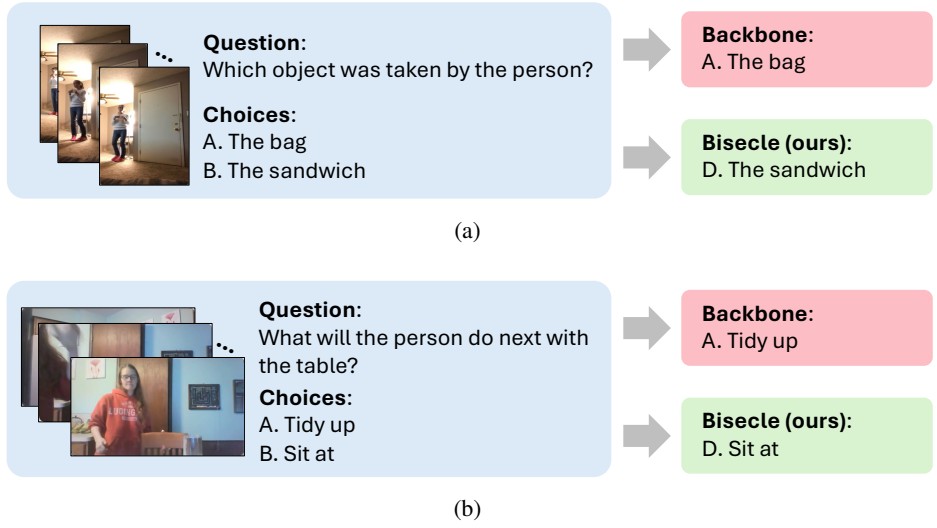

Figure 8: More cases on STAR.

### B.4    Performance of Gemini and GPT

To investigate how frontier multimodal LLMs perform on the tasks in our continual learning sequence, we conduct a series of experiments based on Gemini and GPT family, i.e., Gemini 2.0 Flash, GPT-4o-mini, and GPT-4o. *It is worthwhile to note* that these frontier multimodal LLMs do not support parameter updates, and we can only use APIs to test their performance. Hence, the problems emphasized by this paper, i.e., catastrophic forgetting and update conflict, do not obviously exist for these frontier models and traditional continual learning metrics, e.g., the forgetting rate, cannot be measured appropriately. Although it is unfair for our method that experiences severe forgetting problems during the learning process to compare with these frontier models, this analysis still provides valuable insights into the capabilities of cutting-edge methods on such tasks, highlights their limitations, sheds light on open challenges, and guides subsequent improvements.

**Performance of Gemini and GPT on the video-language tasks.** As shown in Figure 9, we report the test accuracy of each task along the task sequence in NExT-QA dataset. It can be observed that Gemini 2.0 Flash achieves the highest accuracy in almost all tasks. Moreover, although our method experiences the negative impacts brought by continual learning setting, it achieves comparable performance in some tasks (CW, DC, TC, TN, CH) with GPT-4o-mini.

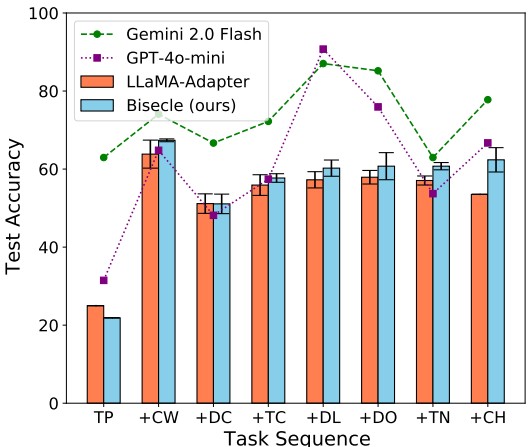

Figure 9: The performance of Gemini 2.0 flash, GPT-4o-mini, LLaMA-Adapter, and our method along the task sequence in the continual learning setting.

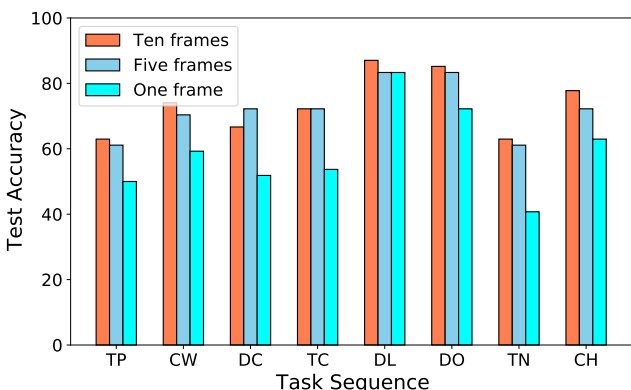

Figure 10: The performance of Gemini 2.0 Flash on different tasks with various numbers of input video frames. The experiments are conducted on NExT-QA dataset.

**Weaknesses of Gemini and GPT.** Despite the strengths of Gemini and GPT, the weaknesses of these frontier MLLMs still exist. *Firstly*, the zero-shot ability of the models highly depends on the number of input video frames. As shown in Figure 10, as the number of video frames decreases,

the performance on all tasks keeps dropping, demonstrating the dependency on sufficient or even redundant visual inputs. *Secondly*, the models have exhibited limited temporal reasoning ability, struggling with tasks that involve delayed causality and action/location sequencing capability. In Figure 11, it can be found that our method outperforms GPT-4o and GPT-4o-mini in task CW and DL, corresponding to "*why do*" and "*where is*", respectively. The former task requires the model to capture cause-and-effect relationships where the effect occurs at a time lag after the cause, rather than instantaneously, while the latter task requires the model to be equipped with sequencing ability, that is to acquire the temporal relationship of different actions and locations. Figure 12 gives a case study of the aforementioned issues faced by frontier models. Specifically, Figure 12(a) shows two failed cases related to delayed causality and Figure 12(b) shows them related to location sequencing.

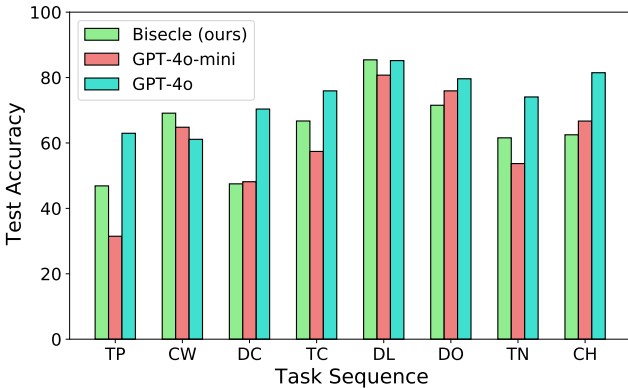

Figure 11: The performance of GPT-4o and GPT-4o-mini on the learning tasks of NExT-QA in our continual learning setting.

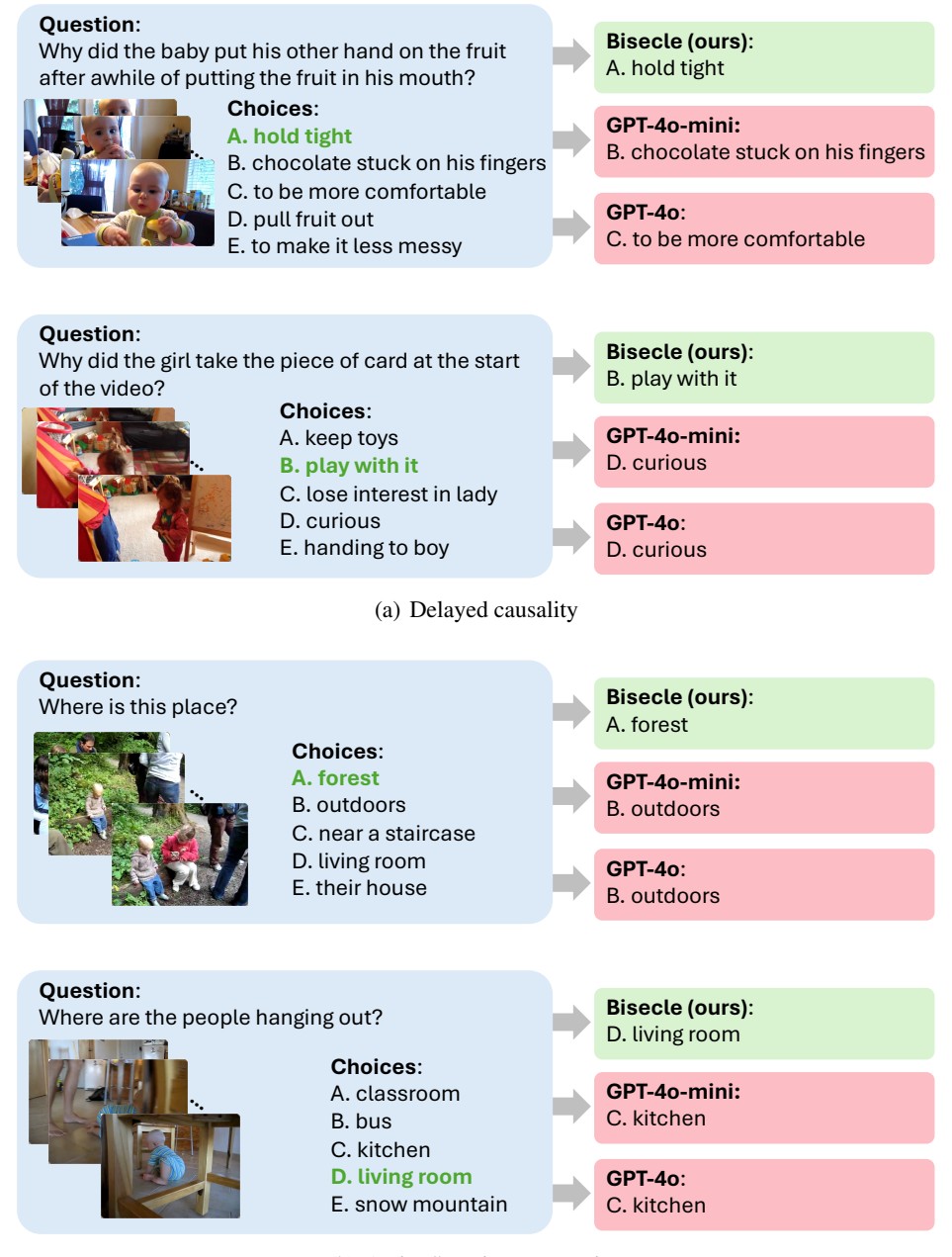

(a) Delayed causality

(b) Action/location sequencing

Figure 12: The weakness of frontier VLMs in dealing with continual learning tasks can be found as limited temporal reasoning ability, which makes them struggle with tasks that involve (a) delayed causality and (b) action/location sequencing.

## B.5 Experiments on Varying LLM Backbones

To examine the generalizability of Bisecle, we further conducted extra experiments on more open-source backbone LLMs, including QWEN-7B and Gemma-7B, with the experimental results shown in Table 11. According to the results, we can see that Bisecle consistently leads to the optimal results compared to the baseline model (i.e., LLaMA-Adapter). This observation illustrates the flexibility and generalizability of our proposed method.

Table 11: The results of LLaMA-A and Bisecle with various LLM backbones.

| Backbones | LLaMA-A | | Bisecle | |
|---|---|---|---|---|
| | Acc | Fog | Acc | Fog |
| LLaMA-2-7B | 46.58 | 13.83 | 62.37 | 5.34 |
| Qwen-7B | 60.77 | 6.54 | 63.97 | 3.57 |
| Gemma-7B | 58.44 | 8.48 | 61.26 | 5.88 |

## B.6 Latency Comparison

We compared the training time of Bisecle and two variants (including the original model), which is shown in Table 12. From the results, we can see that the multi-directional supervision mechanism can lead to longer training time, while contrastive prompt learning only brings minor computational cost. It is reasonable because multi-directional supervision requires end-to-end model training on extra data, leading to increased computational overhead. Compared to the original model, the additional training time is acceptable, while the performance gain is substantial, demonstrating the efficiency of Bisecle.

Table 12: The training time required by Bisecle and two variants.

| $\mathcal{L}_Q$ | $\mathcal{L}_V$ | $\mathcal{L}_P$ | Time | Acc ($\uparrow$) | Fog ($\downarrow$) |
|---|---|---|---|---|---|
| ✗ | ✗ | ✗ | 93 min | 46.58 | 13.83 |
| ✓ | ✓ | ✗ | 171 min | 59.78 | 6.58 |
| ✓ | ✓ | ✓ | 176 min | 62.37 | 5.34 |

# C   Limitations and Broader Impacts

While Bisecle demonstrates significant potential in advancing continual learning for vision-language models through hippocampus-inspired mechanisms, our work primarily focuses on video understanding tasks. Future studies could extend this paradigm to other multimodal domains (e.g., audio-visual learning or embodied AI) and explore its scalability to diverse foundation models. Additionally, the current framework assumes task boundaries are known during training. Relaxing this assumption to enable true task-free continual learning remains an open challenge.

It should be noted that our method is not designed to outperform existing MLLMs, but rather to explore their potential in handling continually evolving data. This work provides preliminary insights for enabling LLMs personalization in dynamic environments. Moreover, given the promising results of binding and separation principles in mitigating forgetting, we believe this work opens new avenues for biologically-inspired learning systems in AI.

