# OpenReview forum: "Bisecle: Binding and Separation in Continual Learning for Video Language Understanding"
_NeurIPS.cc/2025/Conference — NeurIPS 2025 poster_

### Official Review · Reviewer_a9TX · 2025-06-30

**Clarity:** 3
**Significance:** 3
**Originality:** 3
**Rating:** 5
**Confidence:** 4

**Summary:**

Inspired by the neurological principles of how the hippocampus processes complex and dynamic information streams, the authors propose a video continual learning framework for mLLMs. The authors present a cross-modal auxiliary reconstruction task for fast binding across modalities and a task-aware comparative prompt learning to avoid update conflicts between tasks. Experiments demonstrate that the proposed method can effectively improve the generalization and mitigate catastrophic forgetting.

**Questions:**

* How is task type embedding initialized?

**Ethical Concerns:**

["NO or VERY MINOR ethics concerns only"]

**Final Justification:**

I have carefully read the author's rebuttal and other reviewers' comments. I appreciate the authors for providing more implementation details and computational overhead, which makes the article clearer. Overall, my concerns are basically addressed, and I think this paper can provide some inspiration for related areas.

**Limitations:**

Limitations are well discussed in the supplementary material.

**Quality:**

3

**Strengths And Weaknesses:**

**Strengths:**
* The paper is well-motivated, and the components design can be reasonably explained by the principle of the hippocampus.
* The writing is clear, and the paper is well-organized.
* Experiments are comprehensive, validating the effectiveness of the proposed components.

**Weaknesses:**
* Does the training of the Multi-Directional Supervision module need three forward passes? It would be great to show the actual training time or computational overhead compared to the baseline.
* How to compute the contrastive loss when learning the first task? Does this mean that all task settings are pre-defined? It seems to be not reasonable in real-world applications.

---

> ### Author Rebuttal · Authors · 2025-07-30
>
> We appreciate Reviewer a9TX for the valuable feedback and acknowledge our technical contributions and the effectiveness of the proposed method. We address the concerns raised by the reviewer as follows.
>
> **W1:Computational Overhead**
>
> **Answer:** Thanks for your comments. It is true that the multi-directional supervision module requires three forward passes, resulting in a slightly higher training time costs and computational requirements. We provide an additional quantitative experiment on NExT-QA dataset to analyze the time cost in the Table below. It can be observed that compared to the backward propagation, the forward pass incurs much lower computational cost, making it well-suited for multiple executions to enhance model performance.
>
> |L_Q|L_V|L_P|Time|Acc|Fog|
> |-|-|-|-|-|-|
> |✗|✗|✗|93min|46.58|13.83|
> |✓|✓|✗|171min|59.78|6.58|
> |✓|✓|✓|176min|62.37|5.34|
>
> **W2:Calculation of Contrastive loss**
>
> **Answer:** We appreciate your comments. The contrastive loss is mainly discussed in the last paragraph of Sec. 3.3. As shown in Eq. (4), for a reweighted prompt vector computed based on the question token matrix and task-aware prompt matrix, its positive sample is the current task type embedding, while its negative samples include both the previous and current task type embeddings. For the first task, the contrastive loss value equals zero because there are no actual negative samples. By defining the contrastive loss in this way, 1) the loss computation is compatible with the first task, 2) it supports dynamically evolving task streams without the need to pre-define the task sequence.
>
> We are sorry for not making it clear in our current version. We will add related discussion to Sec. 3.3.
>
>
> **Q1:How is task type embedding initialized**
>
> **Answer:** Thanks for your comments. Since the task type embeddings themselves are learnable parameters (line 243), we directly use random initialization to initialize them, similar to other parameters in neural networks. We added the implementation details about how the task type embeddings are initialized in Sec. 3.3 in our future revision.

---

> > ### Comment · Reviewer_a9TX · 2025-08-05
> >
> > Thanks to the author's rebuttal, most of my concerns have been addressed. However, it can be seen that the proposed Multi-Directional Supervision still resulted in nearly double the training cost, which can be further optimized in subsequent works. Overall, I decide to keep my score.

---

> > > ### Author Response · Authors · 2025-08-06
> > >
> > > Dear Reviewer a9TX,
> > >
> > > Thank you for your continued feedback. We appreciate your acknowledgment that your concerns have been addressed. We will incorporate the suggestion into our future work and final revision.
> > >
> > > Authors

---

### Official Review · Reviewer_9h35 · 2025-07-01

**Clarity:** 4
**Significance:** 3
**Originality:** 3
**Rating:** 5
**Confidence:** 3

**Summary:**

This paper is addressing a growing need to extend VLMs to continual video-language tasks.

Its neurobiological framing (binding/separation) is refreshing and clearly influences the algorithm design. The methodology is sound and clearly described, and the experiments are thorough.


However, the novelty is incremental in some respects (prompt tuning + auxiliary objectives), though the grounding in biological metaphors adds conceptual depth.

**Questions:**

1. Can we add more points to Table 4 to see the relationship between #layers and acc?

2. What is the latency, compared to the original model? Need a table comparison

3. Can this be applied to MoE LLMs?

4. Could you provide experiments with QWEN, which will make the proposed method more general?

**Ethical Concerns:**

["NO or VERY MINOR ethics concerns only"]

**Final Justification:**

Based on my suggestions, the authors conducted more experiments, which seems very promising.

**Limitations:**

Yes.

**Quality:**

3

**Strengths And Weaknesses:**

Strengths

1. Very easy to follow
2. Outperforms baselines on NExT-QA, DramaQA, and STAR datasets, improving accuracy by +15.79% and reducing forgetting by 8.49% on NExT-QA
3. The biological analogy with hippocampal mechanisms is more than illustrative—it clearly informs the system design and is empirically validated.

Weaknesses

1. The contrastive learning formulation (Equation 4) assumes well-separated task embeddings, but its sensitivity or failure modes are not discussed.

2. Bisecle still requires prompt injection at up to 32 layers which is super high.

---

> ### Author Rebuttal · Authors · 2025-07-30
>
> We appreciate Reviewer 9h35 for the positive review and constructive comments. We provide our responses as follows.
>
> **W1: Sensitivity or Failure Modes of Contrastive Learning**
>
> **Answer:** Thanks for your valuable question. Since the task embeddings are fully learnable, the inner mechanism of contrastive loss (i.e., closing the positive pairs and separating the negative pairs) can regularize each task embedding to be close to the corresponding task-related query while pushing it away from unrelated task queries, thus promoting better discrimination between different task embeddings. Fig.6 in the supplemental material can prove such separation. This mechanism can mitigate potential failure modes of this loss, such as collapsed or entangled task embeddings. Moreover, Eq. (4) is an additional regularization loss in Bisecle and will not directly control the final prediction of the model. In this case, even if the contrastive loss degenerates into failure modes, the worst-case scenario is that the contrastive mechanism becomes ineffective, without causing a significant negative impact on the final model performance (e.g., worse than the original model).
>
> **W2: Requirements of Prompt Injection Layers**
>
> **Answer:** We appreciate the reviewer for the comment. As discussed in our sensitivity experiments (Line 341-349), there is a trade-off between injected layers and final performance. As shown in Table 4 and Answer to Q1, 12&16 injected layers can also result in competitive performance that exceeds all baselines. Meanwhile, even with 32 injected layers, the number of overall learnable parameters is still far less than the LLM backbone (see Table 3), which demonstrates the parameter efficiency of Bisecle from a marco perspective. We hope our explanation can address your concern.
>
> **Q1: More Points to Table 4**
>
> **Answer:** We thank the reviewer for the valuable comment. As a response, we add more points (12, 20, and 28 layers) to Table 4, and the added results are shown as follows. As we can see in the results, a general trend is that more injected layers can lead to higher accuracy and lower forgetting rates. When the layer number is larger than or equal to 12, the performance becomes competitive, which demonstrates the parameter efficiency of Bisecle.
>
> | Layer | Acc   | Fog   |
> |-------|-------|-------|
> | 8     | 53.86 | 10.13 |
> | 12    | 58.94 | 7.84  |
> | 16    | 58.18 | 7.52  |
> | 20    | 59.23 | 6.96  |
> | 24    | 57.30 | 7.99  |
> | 28    | 58.64 | 6.27  |
> | 32    | 62.37 | 5.34  |
>
>
>
> **Q2: Latency Comparsion**
>
> **Answer:** We appreciate the reviewer for the suggestion. According to your advice, we compared the training time of Bisecle and two variants (including the original model), which is shown in the table below. From the results, we can see that the multi-directional supervision mechanism can lead to longer training time, while contrastive prompt learning only brings minor computational cost. It is reasonable because multi-directional supervision requires end-to-end model training on extra data, leading to increased computational overhead. Compared to the original model, the additional training time is acceptable, while the performance gain is substantial, demonstrating the efficiency of Bisecle.
>
> |L_Q|L_V|L_P|Time|Acc|Fog|
> |-|-|-|-|-|-|
> |✗|✗|✗|93min|46.58|13.83|
> |✓|✓|✗|171min|59.78|6.58|
> |✓|✓|✓|176min|62.37|5.34|
>
> **Q3: Application to MoE LLMs**
>
> **Answer:** Thanks for your inspiring question. While Bisecle can be a plug-and-play module for normal LLMs, it may require several special designs to be adapted to MoE LLMs. The difficulties can lie in how to insert adapter modules into sparsely activated experts without breaking the gating mechanism, and how to ensure efficiency and stability under expert parallelism. Fortunately, several novel approaches, such as MoRA and DynAdapter, have shed light on the application of adapter-like techniques in sparse and expert-based architectures. In this case, we believe that Bisecle can also work with MoE LLMs with minor modification, which can be one of our future directions.
>
> **Q4: Experiments on more LLM Backbones**
>
> **Answer:** Thanks for your comments. To examine the generalizability of Bisecle, we further conducted extra experiments on more open-source backbone LLMs, including QWEN-7B and Gemma-7B, with the experimental results shown below. According to the results, we can see that Bisecle consistently leads to the optimal results compared to the baseline model (i.e., LLaMA-Adapter). This observation illustrates the flexibility and generalizability of our proposed method.
>
> |Method|Acc@LLaMA-2-7B|Fog@LLaMA-2-7B|Acc@Qwen-7B|Fog@Qwen-7B|Acc@Gemma-7B|Fog@Gemma-7B|
> |-|-|-|-|-|-|-|
> |LLaMA-A|46.58|13.83|60.77|6.54|58.44|8.48|
> |Bisecle|62.37|5.34|63.97|3.57|61.26|5.88|

---

> > ### Comment · Reviewer_9h35 · 2025-08-07
> >
> > Thanks for the hard work. I have raised my score.
> >
> > Looking forward to seeing the code and paper later.

---

> > > ### Author Response · Authors · 2025-08-08
> > > **Response to Reviewer 9h35**
> > >
> > > Thank you for your thoughtful feedback and for raising your score. We truly appreciate your support. We’re looking forward to sharing the final version of the paper and releasing the code soon.

---

### Official Review · Reviewer_tR7C · 2025-07-02

**Clarity:** 4
**Significance:** 2
**Originality:** 2
**Rating:** 3
**Confidence:** 4

**Summary:**

This paper investigates the problems of catastrophic forgetting and update conflicts in continual learning for large Vision-Language Models (VLMs). Inspired by neuroscience concepts, the authors propose a framework named Bisecle. This framework strengthens associations within multimodal information via a "Multi-directional Supervision" module and isolates task knowledge through a "Contrastive Prompt Learning" mechanism. Experimental results on multiple Video Question Answering (VideoQA) benchmarks demonstrate the effectiveness of the proposed method. The paper presents clear reasoning, and the experimental section is relatively detailed. The main contribution lies in the innovative proposal to formalize the concepts of "Binding" and "Separating" into two specific learning objectives and apply them to continual learning tasks for VLMs. Nevertheless, there remain some noteworthy issues regarding the depth of analysis of the core mechanisms and the rigor of the experiments, which impact the reliability of the conclusions.

**Questions:**

See above.

**Ethical Concerns:**

["NO or VERY MINOR ethics concerns only"]

**Final Justification:**

I am grateful for the authors' detailed rebuttal. However, my major concern is not fully addressed, that is, the novelty of this paper, especially compared to ColPro. The authors do not demonstrate a sufficient and significant contribution beyond ColPro. Therefore, I choose to keep my score.

**Limitations:**

Yes,  the authors adequately addressed the limitations and potential negative societal impact of their work.

**Paper Formatting Concerns:**

None.

**Quality:**

3

**Strengths And Weaknesses:**

Strengths
- The writing is clear and easy to follow. The motivation is clearly presented.
- The proposed method demonstrates superiority on three videoQA datasets.

Weaknesses
- I don’t think the Neurobiology-Inspired Problem Reframing can be listed as a core contribution. In the continual learning literature, there are many neuro-inspired works [1-4]. Moreover, there is no obvious evidence that the proposed multi-directional supervision module can mimic the hippocampus’s rapid Binding and pattern separation mechanisms. I acknowledge that the proposed method is interesting, but the neuro-inspired contribution is somewhat not very convincing.
- As another LLM-based video understanding framework with continual learning, the ColPro should be carefully discussed and compared. However, it is unclear how is the proposed method superior to ColPro in terms of methodology. The authors only claim that ‘Different from ColPro, our proposed method Bisecle provides a simpler strategy …’. I suggest the authors elaborate their comparison.
- Ablation studies (Table 2) show that the L_Q loss alone yields a significant performance boost on NExT-QA (accuracy rising from 46.58% to 61.13%). The paper attributes this to "strengthening causal understanding," an explanation which is rather vague. Considering that the experimental tasks were divided by question type (e.g., "why", "how"), the L_Q task (predicting the question itself) might unintentionally teach the model to recognize task meta-information (i.e., identify which category the current question belongs to). This learned "task recognition" capability could be the key to the performance improvement, rather than a more generalized "causal understanding." This raises the question: would it still be effective if tasks were partitioned using other criteria, such as video domain (e.g., sports, cooking)? The lack of discussion on this point is a significant limitation of the paper.
- All experimental results report only the mean value from a single run. The standard deviations or error bars from multiple independent runs (e.g., using different random seeds) is missing. This prevents an assessment of the result's stability and potential for randomness.
- On the DramaQA dataset, Bisecle shows a very marginal accuracy improvement compared to ColPro (71.49% vs. 71.24%). This indicates that the advantages of the method are not equally significant across all scenarios.

[1] Incorporating neuro-inspired adaptability for continual learning in artificial intelligence, Nature Machine Intelligence volume 5, pages1356–1368 (2023)

[2] Brain-inspired replay for continual learning with artificial neural networks, Nature Communications volume 11, Article number: 4069 (2020)

[3] Continual Learning Through Synaptic Intelligence, ICML 2017.

[4] Interactive Continual Learning: Fast and Slow Thinking, CVPR 2024.

[5] NISPA: Neuro-Inspired Stability-Plasticity Adaptation for Continual Learning in Sparse Networks, ICML2022

---

> ### Author Rebuttal · Authors · 2025-07-30
>
> We are grateful to Reviewer tR7C for providing insightful feedback. The detailed responses are provided below.
>
> **W1: Concern of Neurobiological Problem Reframing**
>
> **Answer:** Thank you for the thoughtful feedback. We agree that many prior works are inspired by neuroscience, and our goal is not to position the neurobiological analogy itself as a novel contribution. Rather, our contribution lies in the structured reframing of continual video-language learning through two key hippocampal mechanisms, i.e., rapid binding and pattern separation, which directly motivate the design of our multi-directional supervision and contrastive prompt learning modules. **We take the first step to reframe the MLLM-based continual learning from the perspective of how human's hippocampus works.** While we do not claim a precise biological simulation, these modules embody the functional principles of association formation and representation disentanglement observed in the hippocampus. Similar to how neural networks abstractly draw from biological neurons (despite the significant difference between neural networks and real biological neuron systems), our approach uses these neurobiological insights as motivational analogies to guide architectural choices and learning dynamics.
>
> We will add a discussion to further clarify how our neurobiologically-inspired method makes its unique contribution in the current stage of multimodal LLMs and also include an explanation on the contribution compared with other neuroscience-inspired works listed by the reviewer.
>
> **W2: Difference between Ours and ColPro**
>
> **Answer:** We appreciate this insightful suggestion. The difference between Bisecle (our method) and ColPro is three-fold: 1) Bisecle incorporates a multi-directional supervision module to handle catastrophic forgetting via additional supervision signals. In contrast, ColPro does not have any design to augment the supervision signals. 2) ColPro introduces a series of prompt learning techniques in their architecture design, but none of them is designed to handle the update conflict issue. Differently, we introduce a contrastive prompt learning mechanism to address the update conflict problem, using different motivations and designs compared to ColPro. 3) From the motivation perspective, the key designs in our method are inspired by neurobiological mechanisms, with carefully discussed motivations, biological mimicking, and tailored architectural designs. In contrast, ColPro is mainly based on other heuristic motivations, such as question constraint, knowledge acquisition, and visual temporal awareness.
>
> We will include this analysis and comparison in our future revision to better demonstrate the difference between our method and the key baseline.
>
>
> **W3: Reason for Improvement by $L_Q$**
>
> **Answer:** We appreciate this valuable insight. We would like to clarify that the $L_Q$ loss is not limited to predicting the question type (e.g., "why" or "how"). Instead, it involves reconstructing the full question content, including the subject, predicate, and object, and these elements that are often implicitly or explicitly grounded in both the video and the answer. This requires the model to learn structured correspondences across modalities and to recover event semantics, such as causality or temporal dependencies, rather than relying solely on surface-level task cues. Therefore, the observed performance gain reflects enhanced multimodal reasoning, particularly in aligning causal factors across video and textual modalities.
>
> **W4-1: Single-Run Experiments**
>
> **Answer:** Thanks for your question. We now include error bars (standard deviation over 3 runs with different random seeds) and report the accuracy and forgetting rate in the following table. It can be observed that Bisecle has a relatively low standard deviation in both accuracy and forgetting rate, illustrating its stability and robustness across different runs and low sensitivity to initialization.
>
> We did not report error bars on most experiments for the following two reasons: 1) The reported test accuracy and forgetting rate are already averaged over all previous tasks, which partially reflects the model's stability; 2) Generating error bars would require significant additional computation, which was not feasible due to limited computational resources. Considering its significance, we will report more statistical error bars in our future revision.
>
> |Method|Acc(std)|Fog(std)|
> |-|-|-|
> |Bisecle|61.59(0.68)|5.64(0.30)|
>
> **W4-2: Performance Improvement on DramaQA**
>
> **Answer:** We appreciate the review for pointing out the problem. DramaQA is a relatively mature and saturated dataset, where many recent methods already reach a high performance ceiling. In such scenarios, even a small improvement can be meaningful. Also, the forgetting rate of our method is significantly lower than the baselines on this dataset. Moreover, Bisecle demonstrates gains over strong baselines like ColPro on all datasets. This supports the generalizability and robustness of our method across different video-language reasoning tasks.

---

> > ### Comment · Reviewer_tR7C · 2025-08-04
> > **Comment**
> >
> > I am grateful for the authors' detailed rebuttal. However, my major concern is not fully addressed, that is, the novelty of this paper, especially compared to ColPro. The authors do not demonstrate a sufficient and significant contribution beyond ColPro. Therefore, I choose to keep my score.

---

> > > ### Author Response · Authors · 2025-08-06
> > >
> > > We appreciate your continued engagement. Regarding the novelty of our method (Bisecle) compared to ColPro and the performance gap between the two, we would like to provide further clarification and supporting evidence.
> > >
> > > **Methodological Innovations and Key Differences between Bisecle and ColPro**
> > >
> > > We would like to emphasize that Bisecle and ColPro are fundamentally different in terms of methodology. The only superficial similarity is that both methods utilize learnable prompts and impose constraints on them via a loss function. However, prompt learning has become a common and efficient fine-tuning strategy in the era of large-scale models, and applying a loss to guide prompt learning is a standard practice to optimize them.
> > >
> > > Beyond the above similarities, Bisecle introduces several novel and distinctive methodological components that are not present in ColPro. Specifically:
> > > 1) Bisecle incorporates a multi-directional supervision mechanism inspired by neurobiological processes to better preserve knowledge and separate task-specific representations. This mechanism serves as a key component that contributes to the strong performance of Bisecle by augmenting supervision signals during training. In contrast, ColPro does not incorporate any mechanism for supervision signal enhancement or representation separation from this perspective.
> > > 2) We propose a contrastive prompt learning strategy to explicitly mitigate update conflicts during continual learning. In contrast, ColPro does not include specific mechanisms to address update conflicts.
> > > 3) The two components in Bisecle, inspired by neurobiological principles, are seamlessly integrated to form a unified framework that jointly addresses catastrophic forgetting and update conflict issues. The two components contribute from different yet complementary angles: one strengthens the model via enriched supervision signals, and the other refines the prompt representations to ensure stable and adaptive learning across tasks. In contrast, ColPro lacks such a targeted and comprehensive design, relying instead on heuristic prompt optimization alone.
> > >
> > > To sum up, Bisecle is methodologically distinct from ColPro, both in motivation and design. We kindly ask the reviewer to reconsider the originality and significance of our contributions.
> > >
> > >
> > > **Performance Improvement beyond ColPro**
> > >
> > > In the comments by Reviewer tR7C, we appreciate the reviewer’s attention to the performance comparison in terms of accuracy. Notably, Bisecle consistently outperforms ColPro across all datasets, with significant improvements on NExT-QA (+13.1%) and STAR (+7.2%).
> > >
> > > Beyong accuracy, we believe the **forgetting rate** was overlooked in the evaluation. In continual learning scenarios, mitigating forgetting is often more critical than achieving marginal accuracy gains, as it directly impacts the model's robustness and applicability in real-world deployments. We would like to further highlight the superiority of our method in terms of forgetting mitigation:
> > >
> > > | Dataset | ColPro Fog(↓) | Bisecle Fog(↓) | Relative Reduction |
> > > | ------- | ------------- | -------------- | ------------------ |
> > > | NExT-QA | 7.43%         | 5.34%          | 28.12%             |
> > > | DramaQA | 12.64%        | 10.37%         | 17.96%             |
> > > | STAR    | 8.13%         | 7.60%          | 6.52%              |
> > >
> > > This consistent forgetting reduction across all three datasets demonstrates that Bisecle can retain more knowledge from previous tasks than ColPro. In Continual learning scenarios, a model that forgets 28% less is practically superior for real-world deployment, as it can:
> > > 1) Learn more tasks sequentially without catastrophic forgetting;
> > > 2) Maintain better performance on earlier tasks overtime;
> > > 3) Provide more reliable service in dynamic environment.
> > >
> > > In this case, we believe that Bisecle offers not only stronger empirical results but also better practical utility, highlighting its value beyond accuracy alone.

---

> > > > ### Author Response · Authors · 2025-08-08
> > > >
> > > > Regarding Reviewer tR7C's concerns about our neuro-inspired contribution, we would like to clarify that Bisecle is designed with the goal of drawing inspiration from biological neural mechanisms through well-motivated deep learning architectures and components, in order to achieve more effective continual learning, just like the neuro-inspired works [1-5] listed by the reviewer.
> > > >
> > > > For instance, [4] is inspired by the complementary learning system (CLS) theory in neurocognitive science which consists of fast thinking (System 1) and slow thinking (System 2). As a deep learning implementation, ICL [4] uses a ViT to emulate the fast responses of System 1, and a multimodal LLM to capture the reasoning-based processing of System 2. While the components (e.g., ViT and LLM) may not fully replicate the underlying memory mechanisms, the neurocognitive inspiration still offers valuable guidance for designing effective learning mechanisms in deep models.
> > > >
> > > > Similar to this line of work, the design of Bisecle is also rooted in high-level neurobiological principles, i.e., the memory mechanism of hippocampus. Specifically, we integrate a multi-directional supervision mechanism to reflect the rapid binding and pattern separation functions of the hippocampus, and a contrastive prompt learning module to resolve update conflicts. The two components together enable better knowledge retention and task-level decoupling in continual learning of VLMs. The most significant contribution here lies in the concept-level inspiration, rather than low-level biological mimicry. This rationale is also common in other neuro-inspired works (a comparsion see the table below).
> > > >
> > > > Finally, we would like to note that although our work shares a common neuro-inspired perspective with [1–5], the specific neural mechanisms we draw from, the continual learning scenarios we target, and the design of our method are entirely different. Our novelty has also been acknowledged by other reviewers. Therefore, while the neuro-inspired design patterns in these works have provided us with valuable inspiration (and we will cite and discuss them in the final version), their existence does not diminish the originality of our contributions.
> > > >
> > > > | paper                 | Biological principle cited                               | Engineering abstraction                                      |
> > > > | --------------------- | -------------------------------------------------------- | ------------------------------------------------------------ |
> > > > | [1] | Complex molecular synaptic consolidation                 | Per-synapse importance variable & quadratic penalty          |
> > > > | [4]        | Complementary learning system (fast vs. slow memory)     | ViT(system 1) + multimodal-LLM(system 2) interaction         |
> > > > | [5]                 | Stability-plasticity balance, sparse structural rewiring | Fixed-density sparse net + dynamic rewiring & stable units   |
> > > > | **Ours**     | **Hippocampal rapid binding & pattern separation**       | **Multi-directional supervision(binding) + contrastive prompt learning(separation)** |
> > > >
> > > >
> > > > [1] Incorporating neuro-inspired adaptability for continual learning in artificial intelligence, Nature Machine Intelligence volume 5, pages1356–1368 (2023)
> > > >
> > > > [2] Brain-inspired replay for continual learning with artificial neural networks, Nature Communications volume 11, Article number: 4069 (2020)
> > > >
> > > > [3] Continual Learning Through Synaptic Intelligence, ICML 2017.
> > > >
> > > > [4] Interactive Continual Learning: Fast and Slow Thinking, CVPR 2024.
> > > >
> > > > [5] NISPA: Neuro-Inspired Stability-Plasticity Adaptation for Continual Learning in Sparse Networks, ICML2022

---

### Official Review · Reviewer_EdJu · 2025-07-03

**Clarity:** 3
**Significance:** 3
**Originality:** 3
**Rating:** 4
**Confidence:** 3

**Summary:**

This paper presents Bisecle, a framework for continual learning in video-language understanding, inspired by neurobiological mechanisms, specifically the hippocampus's binding and pattern separation functions. The authors aim to address key challenges in continual learning: catastrophic forgetting and update conflicts, which arise when models try to adapt to new tasks without overwriting previous knowledge. Bisecle integrates a multi-directional supervision module and contrastive prompt learning to mitigate these issues while maintaining parameter efficiency. The paper demonstrates the effectiveness of Bisecle through extensive experiments on several VideoQA benchmarks, showing improvements in task retention and cross-task generalization.

**Questions:**

1 Error Bars and Statistical Significance: The results would be much more convincing if the paper included error bars or some statistical analysis to quantify the uncertainty in the reported performance. Would the authors consider providing these or explaining the rationale for omitting them?
2 Clarification of Mechanisms: The contrastive prompt learning and multi-directional supervision mechanisms are innovative, but the paper could benefit from further clarity on how these mechanisms scale with different model sizes or task complexities. Could the authors provide more insights into the scalability of Bisecle for larger or more diverse tasks?
3 Computational Efficiency: While the paper emphasizes parameter efficiency, can the authors provide more specific benchmarks on how Bisecle compares in terms of computational cost, especially when scaled up to larger models or more complex datasets? It would be helpful to understand the trade-offs between computational cost and performance improvements.

**Ethical Concerns:**

["NO or VERY MINOR ethics concerns only"]

**Final Justification:**

The clarification on computational efficiency and the inclusion of error bars are helpful. It’s good to see the explanation regarding the scalability of Bisecle and the additional time for multi-directional supervision. The benchmarks on performance and computational cost are also appreciated. All my questions are clarified, so I decide to maintain my positive score.

**Limitations:**

yes

**Paper Formatting Concerns:**

Consistent Use of Bold and Italics: In the text, the use of bold and italics appears somewhat inconsistent. Double-check that their usage is in line with the standard formatting practices for emphasis or notation.

**Quality:**

3

**Strengths And Weaknesses:**

Strengths:
1. Novelty: This paper offers a fresh approach inspired by neurobiology, specifically the hippocampus's memory mechanisms, which is uncommon in video-language models. The use of multi-directional supervision and contrastive prompt learning creatively tackles catastrophic forgetting in large multimodal models, providing a unique solution to memory retention in dynamic data streams.

2. Significance: The method addresses real-world challenges in video-language understanding, particularly for continuous data like dynamic scene analysis from wearable devices. It bridges cognitive neuroscience and machine learning, with strong empirical results showing practical impact on VideoQA benchmarks.

3. Technical Rigor: The methodology is clear and well-justified, with robust experiments comparing Bisecle to existing approaches. The results demonstrate notable improvements in reducing forgetting and enhancing cross-task generalization.

Weaknesses
Complexity of Mechanisms: While the neurobiological inspiration is compelling, the methods, particularly the contrastive prompt learning, may be seen as complex. Although the paper claims computational efficiency, a deeper discussion of the trade-offs and overhead introduced by the task-specific prompts would be valuable.

---

> ### Author Rebuttal · Authors · 2025-07-30
>
> We appreciate Reviewer EdJu for the perception of our contributions and thank the reviewer for the insightful feedback. The detailed responses are provided below.
>
> **W1. Complexity of Mechanisms**
>
> **Answer:** We appreciate your point regarding the overhead of Bisecle, especially contrastive prompt learning. We would like to verify its computational efficiency from two aspects. 1) Complexity analysis. Given the contrastive module, the time complexity for each sample is $O(NDT + DT^2)$ (detailed analysis will be provided in our future version), where $N$ is the token number, $D$ is the latent dimension, and $T$ is the task number. Since $T$ is usually far less than $N$ and $D$, while the cost is linear in the sequence length, the overall computation cost of the contrastive module is relatively low and scalable in practice. 2) As shown in the table in the answer to Q1, the contrastive learning module only brings 2.9% (5 min of 171 min) extra time, which confirms its practical efficiency and minimal cost.
>
> **Q1. Error Bars and Statistical Significance**
>
> **Answer:** Thanks for your question. We now include error bars (standard deviation over 3 runs with different random seeds) and report the accuracy and forgetting rate in the following table. It can be observed that Bisecle has a relatively low standard deviation in both accuracy and forgetting rate, illustrating its stability and robustness across different runs and low sensitivity to initialization.
>
> We did not report error bars on most experiments for the following two reasons: 1) The reported test accuracy and forgetting rate are already averaged over all previous tasks, which partially reflects the model's stability; 2) Generating error bars would require significant additional computation, which was not feasible due to limited computational resources. Considering its significance, we will report more statistical error bars in our future revision.
>
> |Method|Acc(std)|Fog(std)|
> |-|-|-|
> |Bisecle|61.59(0.68)|5.64(0.30)|
>
>
> **Q2: Clarification of Mechanisms**
>
> **Answer:** Thank you for the insightful question. In terms of model size, the results in Table 3 show that Bisecle is compatible with LLMs ranging from 1B to 13B, introducing only a small number of additional parameters and computational cost. In terms of task complexity, it is usually correlated with the length of videos and the length of texts, which are both approximately proportional to the number of tokens. As the complexity of all modules in Bisecle is typically either independent of or linear with respect to this number, the additional computational cost introduced by task complexity is aligned with that of the transformer-based LLM itself. To sum up, Bisecle enjoys comparable scalability to the LLM backbone with respect to both model sizes and task complexities.
>
> **Q3: Computational Efficiency**
>
> **Answer:** We appreciate the reviewer for the suggestion. According to your advice, we compared the training time cost of Bisecle and two variants (including the original model), which is shown in the table below. From the results, we can see that the multi-directional supervision mechanism can lead to longer training time, while contrastive prompt learning only brings minor computational cost. It is reasonable because multi-directional supervision requires end-to-end model training on extra data, leading to increased computational overhead. Compared to the original model, the additional training time is acceptable (<2x), while the performance gain is substantial (33.8%), demonstrating the efficiency of Bisecle. Due to the limitations of time and space, we did not report the running time for more baselines and datasets, and these will be provided in our future revision.
>
> |L_Q|L_V|L_P|Time|Acc|Fog|
> |-|-|-|-|-|-|
> |✗|✗|✗|93min|46.58|13.83|
> |✓|✓|✗|171min|59.78|6.58|
> |✓|✓|✓|176min|62.37|5.34|
>
> **C1: Consistent Use of Bold and Italics**
>
> **Answer:** Thank you for pointing this out. We will correct and standardize all bold/italic usages in the final version.

---

> > ### Comment · Reviewer_EdJu · 2025-08-06
> >
> > Thank you for the detailed responses to the comments. The clarification on computational efficiency and the inclusion of error bars are helpful. It’s good to see the explanation regarding the scalability of Bisecle and the additional time for multi-directional supervision. The benchmarks on performance and computational cost are also appreciated.

---

### Note · Authors · 2025-08-15

**Dear Area Chair and Reviewers,**

We sincerely thank the area chair for your exceptional leadership throughout this review process, and all reviewers for your insightful feedback. Your time and expertise have been invaluable in improving our work, and the careful consideration given to our research is deeply appreciated.

Regarding the rebuttal process, we are pleased to report that the majority of participating reviewers have acknowledged our comprehensive responses to their concerns, and have accordingly **voted in favor of accepting our manuscript**:

`Reviewer EdJu` expressed satisfaction with our clarification on computational efficiency, the inclusion of error bars, and the explanation regarding the scalability and time consuming of Bisecle.

`Reviewer 9h35` appreciated our additional experiments where a latency comparsion of running time, detailed parameter analysis, and genralization analysis on extra LLM backbones (i.e., Qwen and Gemma) are provided, which prompted the reviewer to raise the score.

`Reviewer a9TX` indicated approval of our expanded discussion of methodology details of contrastive loss and embedding initialization.

In response to the concerns raised by `Reviewer tR7C`, we applied the same level of careful attention and thoroughness. First, we offered supplementary experiments and justifications repsonding to each concern raised. Additionally, we provided a detailed follow-up response specifically addressing their concerns on the difference between our method and the baseline ColPro. Moreover, we compared our method with other neuro-inspired studies to demonstrate our neurobiological contribution and clarified the originality of our neuro-inspired design.

Overall, we believe the review process has provided a valuable opportunity to strengthen our work and address the reviewers' insightful comments. While we are pleased to have resolved the majority of concerns raised, we remain hopeful that the efforts we’ve made to refine our work will be thoroughly considered in your final evaluation.

Once again, we extend our sincere appreciation for your guidance and thoughtful consideration of our manuscript. We greatly value the AC's leadership and look forward to your fair and balanced evaluation of our work.


Best regards,

Authors

---

### Decision · Program_Chairs · 2025-09-17

**Decision:**

Accept (poster)

**Comment:**

This paper presents Bisecle, a neuro‑inspired continual learning framework for video‑language understanding that combines multi‑directional supervision and contrastive prompt learning to address catastrophic forgetting and update conflicts, achieving improved task retention and cross‑task generalization on multiple VideoQA benchmarks with parameter‑efficient adaptation.

Strengths:
- Clear motivation and presentation: Multiple reviewers highlight the clarity, organization, and ease of following the paper’s narrative and problem framing.
- Neurobiology‑inspired design informing the method: The work offers a fresh approach inspired by hippocampal memory mechanisms, which is uncommon in video‑language models. The integration of multi‑directional supervision and contrastive prompt learning creatively addresses catastrophic forgetting in large multimodal models, providing a distinctive solution to memory retention in dynamic data streams.
- Methodological soundness with targeted objectives: The combination of multi‑directional supervision (knowledge preservation) and contrastive prompt learning (reducing update conflicts) is well‑justified and effective under parameter‑efficient adaptation.
- Strong empirical results across datasets: Consistent gains over baselines on NExT‑QA, DramaQA, and STAR, including notable improvements in accuracy and reductions in forgetting (e.g., on NExT‑QA: +15.79% accuracy, −8.49% forgetting).

Weaknesses:
- The key unresolved concern is that the contribution beyond ColPro is not sufficiently demonstrated; one reviewer maintains a novelty objection despite the rebuttal.
-  The proposed multi‑directional supervision still results in nearly double the training cost, which could be further optimized in future work.

Final Recommendation:

The authors provided additional experiments (e.g., error bars, latency comparison, Qwen/Gemma backbones) and clarifications in rebuttal that addressed the majority of reviewer questions. Three of four reviewers now lean positive, citing strong empirical results and methodological soundness. I recommend acceptance, but note that the authors should include in the final version the additional experimental results they committed to (e.g., running time for more baselines and datasets).